# Virtual Golf, “Exergaming”, Using Virtual Reality for Healthcare in Older Adults: Focusing on Leisure Constraints, Participation Benefits, and Continuous Participation Intention

**DOI:** 10.3390/healthcare12100962

**Published:** 2024-05-08

**Authors:** Chulhwan Choi, Dong-Kyu Kim, Inyup Lee

**Affiliations:** 1Department of Physical Education, Gachon University, 1342, Seongnam-daero, Sujeong-gu, Seongnam-si 13120, Republic of Korea; chulhwanchoi@gachon.ac.kr; 2Department of Sport, Leisure & Recreation, College of Nature Science, Soon Chun Hyang University, Ansan-si 31538, Republic of Korea; 3Department of Sport Science, Jeju National University, 102, Jejudaehak-ro, Jeju-si 63243, Republic of Korea

**Keywords:** exergaming, virtual reality, older adults, constraints, benefits, continuous participation

## Abstract

Exergaming, a new type of sport, combined with virtual reality, has provided new opportunities for the aging population. This study analyzed the differences in leisure constraints, participation benefits, and continuous participation intention in virtual golf (represented as an exergame) depending on the participants’ ages. Data collection was conducted from August 2023 to November 2023. A quantitative research design and a convenience sampling method were employed, targeting 310 regular virtual golf participants aged 20 years or older in the Republic of Korea. For comparative analysis, the survey participants were segmented into three groups: Group 1, young adults (18–35 years); Group 2, middle-aged adults (36–55 years); and Group 3, older adults (56–69 years). To compare and analyze participation behaviors in virtual golf, the dependent variables were identified: (a) leisure constraints (four factors) to limit formation and participation in leisure; (b) participation benefits (four factors) to encourage participation in leisure; and (c) continuous participation intention (single factor) to show likelihood to participate in leisure in the future. The results revealed that the young adult group showed statistically high results for costs under leisure constraints (*F* = 14.949, *p* < 0.001, *η_p_*^2^ = 0.089), and the older adult group reported statistically high results in physical (*F* = 9.346, *p* < 0.001, *η_p_*^2^ = 0.057) and mental (*F* = 7.249, *p* < 0.001, *η_p_*^2^ = 0.045) participation benefits and continuous participation intention (*F* = 6.486, *p* < 0.01, *η_p_*^2^ = 0.041). This study confirmed that virtual golf using advanced technology brings physical and mental benefits to older people based on reasonable cost and enables continuous participation in physical activity.

## 1. Introduction

According to the Korean Statistical Information Service [1], the median age of the total Korean population is expected to increase from 42.1 in 2019 to 46.7 in 2030 and exceed 50 in 2040. The United Nations (UN) defines an “aging society” as one in which the proportion of people aged 65 years and older accounts for 7% or more of the total population, an “older society” for 14% or more, and an “ultra-older society” for 20% or more [2]. Based on these criteria, South Korea has already become an aging society, with 14% of the population aged 65 years and older. According to the theory of successful aging [3], older adults who enjoy healthy physical activity are more likely to live healthy lives and have better mental health. Sirven and Debrand [4] studied middle-aged and older adults aged 50 years and older in 11 European countries and found that those who were more physically active reported higher subjective health status. This suggests that commitment to healthy living increases with age. Furthermore, similar studies [5,6] showed that relatively young people under the age of 30 years had lower interest and participation rates in physical exercise.

The first benefit achieved when older adults participate in physical activity is physical benefit. While it is not possible to completely prevent the decline in physical fitness in older adults due to aging, regular exercise and appropriate physical activity can not only slow down the rate of decline but also increase it [7]. Physical fitness not only helps maintain or improve health, but it is also essential for performing daily activities. Regular physical activity can reduce the incidence and risk factors for chronic diseases such as hypertension and obesity, which can be associated with physical inactivity due to aging [8]. In addition to physical health, mental health is an important aspect of physical activity. Participation in physical activity can reduce depression and promote a positive psychological state, which is beneficial for psychological health [9]. Appropriate physical activity can also help address negative emotions such as feelings of isolation and loneliness that are common among older adults. In addition, physical activity in older adults affects social relationships [10], and it has been shown that participation in physical activity in older adults can promote social health through bonding and interaction and has a substantial impact on mental health, including stress relief and psychological well-being [11,12]. Therefore, it is necessary to expand efforts to plan and provide exercise programs for older adults in the community to increase and sustain their participation in health exercises. It is important to improve the quality of life of older adults through exercise or physical activity so that they can perform daily activities independently and live a happy and healthy old age [13]. 

Over the past three years, the movement toward games that incorporate virtual reality (VR) has grown in the wake of the coronavirus disease 2019 (COVID-19) pandemic [14]. VR technology is the virtual representation of something that does not exist in the real world, and its introduction to leisure activities is revolutionary [15]. Exergaming is a new concept that combines exercise and gaming, in which the act of playing a game is itself an exercise [16]. Exergaming is applied in many areas, including prevention, treatment, and rehabilitation, and it reflects the diversity of research fields [17]. Moreover, this diversity focuses on different age groups, ranging from children to older adults [18]. As it is difficult for older people to perform strenuous exercises that require considerable movement, exergaming can be optimized in a manner that allows them to play games in VR with relative ease [19]. Compared to traditional exercise programs, exergaming has the advantage of being based on games and fun, which can help people exercise longer, more often, and more regularly [20].

Virtual golf is a virtual sport that allows users to enjoy a virtual space through 3D modeling of real golf courses. Specifically, players hit the real ball with real golf clubs in front of a large projected screen realized via a simulator. The simulation system provides VR based on club head speed, the angle of the club face, the spinning rate, direction, distance, and realistic visuals of the landscape, slope, fairway, rough, and sand bunker [21,22]. Additionally, the basic posture, rules, and field environment related to golf can be learned by beginners in virtual settings, much like a real golf course [23]. Furthermore, virtual golf has been shown to solve real golf problems. Lee and Jee [24] found that participants were enthusiastic about the emergence of virtual golf due to its high geographical accessibility, ease of participation in various settings, and price competitiveness due to its low cost. Virtual golf has made it possible to enjoy a VR experience that closely resembles an actual golf game, and it is expected that it will be possible to enjoy various leisure activities other than golf in a VR space in the future. In conclusion, virtual golf can improve the quality of life of older adults individually and play an important role in national health insurance finances [25]. Studies [9,12,13,14,15] have been conducted on healthy aging and physical activity; while some have applied VR, most have focused on therapeutic or rehabilitative applications. Given that, this study could provide a scientific approach to analyzing whether a new type of sport with advanced technology can play an important role in the aging era by conducting a study on exergaming with VR that can be applied to promote the health of older people in everyday life.

Hence, this study aims to empirically analyze whether differences exist in leisure constraints, participation benefits, and continuous participation intention in virtual golf, which can be represented as exergaming, depending on the participants’ age. The hypotheses established in this study to scientifically prove the effect of technological development on the improvement of physical activity in older people are as follows:

**Hypothesis 1.** 
*Participation constraints differ according to the age of virtual golf participants.*


**Hypothesis 2.** 
*Participation benefits differ based on the age of virtual golf participants.*


**Hypothesis 3.** 
*Continuous participation intention differs based on the age of virtual golf participants.*


## 2. Materials and Methods

### 2.1. Study Design

This study investigated the differences in participation constraints, participation benefits, and continuous participation intention in virtual golf among three age groups to verify the possibility that a new type of sport with the latest technology can have a positive impact on older adults. Furthermore, this study applied a quantitative research design and used convenience sampling. Online and offline survey questionnaires were given to the participants stating the purpose of the study and instructions for study participation. The collection process took place over four months starting in August 2023.

### 2.2. Participants

The survey respondents were limited to adults aged 20 years or older in the Republic of Korea who regularly participated in virtual golf over the past year (at least four times a month). Conversely, subjects were excluded from this study if they were minors aged under 20 years or did not regularly participate in virtual golf. Each survey respondent who voluntarily participated was briefly informed about the study’s purpose. An a priori power analysis was conducted using G*Power version 3.1.9.7 for sample size estimation with a significance criterion of *α* = 0.05 and power = 0.95. The result showed that the minimum sample size needed with a large effect size was 246 for this study.

### 2.3. Data-Collection Tools

Age was used as an independent variable for comparative analysis. Based on the collected age information, the survey respondents were divided into three groups: Group 1, young adults (18–35 years); Group 2, middle-aged adults (36–55 years); and Group 3, older adults (56–69 years). In this study, the factors used to measure leisure constraints and the continuous participation intention of survey respondents were applied to questionnaires modified by Choi [23], who investigated consumer behaviors toward virtual golf from a marketing perspective. The participation constraint factor consists of four sub-factors: (a) cost (three items), (b) health (three items), (c) confidence (three items), and (d) social (three items). Continuous participation intention was a single-scale factor that included three items. Next, the factor used to understand the benefits of participation in virtual golf in this study was applied to questionnaires modified by Bum, Yang, and Choi [26], exploring differences between actual and virtual golf. Participation benefits also had four sub-factors: (a) personal satisfaction (three items), (b) physical benefits (three items), (c) social benefits (three items), and (d) mental benefits (three items). All survey questionnaire items utilized a 5-point Likert scale ranging from 1 (“not at all”) to 5 (“very much”).

### 2.4. Data Analysis

The collected survey data were statistically analyzed using SPSS version 28.0. Descriptive statistics, including sociodemographic data, were analyzed. Next, scale validity was verified through exploratory factor analyses (EFA), and scale reliability was tested using Cronbach’s alpha coefficients for three factors (i.e., leisure constraints, participation benefits, and continuous participation intention). Finally, multivariate analysis of variance (MANOVA), including a post-hoc analysis, was performed to determine the statistical differences in variables among the three groups segmented based on age. MANOVA is a statistical method that has been widely used [26] in the social sciences to compare the mean values of dependent variables among groups segmented by independent variables. The significance threshold was set at *p* = 0.001.

## 3. Results

### 3.1. Descriptive Statistics

Data were collected from 310 survey questionnaires. As mentioned above, the minimum sample size needed with a large effect size was 246 based on the result of the a priori power analysis using G*Power version 3.1.9.7. Thus, to verify the research hypotheses, the obtained sample size of 310 was satisfactory. Detailed descriptive statistics, including sociodemographic information of the survey respondents, are presented in Table 1.

### 3.2. Scale Validity and Reliability

The two EFAs with principal component analysis (PCA) were implemented separately for each factor: (a) participation constraints (cost (three items), health (three items), confidence (three items), and social (three items)) and (b) participation benefits (personal satisfaction (three items), physical (three items), social (three items), and mental (three items)). Continuous participation intention as a single-scale factor including three items (“I will continue to participate in virtual golf”, “I would recommend virtual golf to my friends and family”, and “I will purchase equipment to continue participating in virtual golf”) was excluded from factor analysis. First, in terms of the factor structure of participation constraints, the Kaiser–Meyer–Olkin test showed sample adequacy (0.727) [27]. In addition, Bartlett’s test of sphericity found statistical significance (χ^2^ = 1892.888, *df* = 66, *p* < 0.01). This EFA retained four factors (cost, health, confidence, and social), explaining 78.225% of the total variance with satisfactory statistical standards (greater than one eigenvalue and greater than 0.40 factor structure coefficients). In addition, the factors met acceptable internal consistency based on the results from Cronbach’s alpha coefficients: cost (*α* = 0.868), health (*α* = 0.849), confidence (*α* = 0.857), and social (*α* = 0.841) [28] as shown in Table 2.

In terms of the factor structure of participation benefits, the Kaiser–Meyer–Olkin test showed sample adequacy (0.761) [27]. Additionally, Bartlett’s test of sphericity satisfied statistical significance (χ^2^ = 2062.611, *df* = 66, *p* < 0.01). The remaining four factors (personal satisfaction, physical, social, and mental) explained 78.455% of the total variance and had acceptable eigenvalues and factor structure coefficients. 

For internal consistency, all factors had acceptable (greater than 0.70) Cronbach’s alpha coefficients: personal satisfaction (*α* = 0.894), physical (*α* = 0.856), social (*α* = 0.840), and mental (*α* = 0.833) [28] (Table 2). Finally, the factor of continuous participation intention (*α* = 0.850) excluded from factor analysis had acceptable reliability.

### 3.3. Multivariate Analysis of Variance

Before the analysis, assumptions were checked: (a) independence, (b) random sampling, and (c) homogeneity of covariance metrices. In this study, the randomly sampled respondents completed the survey independently without duplication. In addition, to verify the assumption of homogeneity of covariance metrices, Box’s test was performed; this study had three groups of fairly equal sample sizes, which indicated that a group should be no larger than 1.5 the size of another group. 

A MANOVA was applied to verify differences in leisure constraints, participation benefits, and continuous participation intentions. First, the homogeneity of covariance was analyzed (Box’s *M* = 199.821, *F* = 2.127, *p* < 0.01). In addition, statistically significant differences among the three groups were revealed (Wilks’ Lambda = 0.759, *F* = 4.919, *p* < 0.01, *η_p_*^2^ = 0.129). Specifically, the analysis reported statistically significant mean differences in (a) the cost of participation constraints, (b) physical participation benefits, (c) mental participation benefits, and (d) continuous participation intention. 

To determine which of the three groups showed significant differences, a post hoc analysis was performed. First, in terms of cost of participation constraints, the young adult group (G1) had relatively higher scores than the middle-aged adult (G2) and older-aged adult (G3) groups. Regarding the benefits of physical participation, the older adult group (G3) showed higher scores than the middle-aged group (G2). Third, regarding mental participation benefits, the older adult group (G3) generated higher scores than the young adult group (G1) and the middle-aged group (G2). Finally, regarding continuous participation intention, higher average scores were obtained for the older adult group (G3) than for the young adult group (G1). Detailed results of the multivariate analysis of variance (Table 3) and post-hoc analyses (Table 4) are reported below.

## 4. Discussion

This study analyzed how virtual golf, a representative type of exergaming, provides leisure participation experiences by age. The younger age group experienced participation constraints in terms of participation costs, whereas the older age group experienced participation benefits in terms of physical and mental aspects. In addition, the intention to continue participation, which is considered the most important aspect of leisure participation, was significantly higher in older age groups. This study provides objective evidence that virtual golf, as an exergame, can provide meaningful value to the aging population of the leisure industry. Detailed interpretations for each statistically significant result are provided below.

First, the participants in this study showed high results for the cost factor among leisure participation constraints in the relatively younger age group who participated in virtual golf. Previous studies [25,26] have found statistically significant results for the cost of participation, which has been analyzed as an advantage of virtual golf. Virtual golf provides a reasonable participation cost, which can be a burden for participants of actual golf [26]. However, if participants are to be divided into age groups and their perceptions are compared, as in the design of this study, younger participants are expected to find it relatively costly. Research results have been reported that young leisure participants are changing their leisure choices due to the recent rapidly changing economic situation [29]. In particular, the outbreak and end of the coronavirus led to this change [30]. This study analyzed the expansion of the older adult population and identified a new type of sport that can appeal to them, particularly to female older adults. In general, economic changes after retirement limit leisure participation among older adults [17,18]. In the case of virtual golf, the financial burden was lower for older participants than for participants in other age groups. It is hoped that virtual golf will have a positive effect on the leisure sports industry by providing more opportunities for older leisure participants in terms of cost. Furthermore, the frequency of virtual golf participation by age group and participation patterns should be analyzed. 

The next statistically significant finding of this study was the physical and mental benefits of leisure participation. Specifically, the results were higher in the older group (group 3). This suggests that older virtual golf participants were more likely than younger participants to experience physical and mental benefits from participating in virtual golf. It is worth paying attention to studies that have argued that just being able to get opportunities for leisure participation can benefit the older adults [31]. Our findings have several important implications. Previous studies have highlighted the physical, mental, and social benefits of participation in leisure sports [26,32,33,34]. In addition, the importance of physical activity has been emphasized since the COVID-19 pandemic and government policies are changing to reflect its importance [35]. The fact that the findings of this study were replicated in a new type of virtual golf, one of the most advanced forms of exergaming, provides objective data that can be used to inform future initiatives to engage older adults. In other words, not only traditional leisure sports but also exergaming can play a role in leisure sports. The importance of this study is further emphasized when considering the results of previous studies showing that virtual golf reduces the participation constraints of participants compared to real golf [25]. By reducing the burden and increasing the benefits, virtual golf could be a new solution for the older adult population, which is considered a marginalized group in the leisure industry. This accessible form of exergaming could be further expanded and developed to allow more people to experience the benefits of leisure participation. Furthermore, based on the results of this study, it is possible to experience the psychological benefits of virtual golf beyond its more commonly recognized physical benefits.

Finally, among the dependent variables that were compared and analyzed to obtain statistically significant results, continuous participation intention was the most significant. The results of this study indicate that older adult participants show stronger participation intentions. Previous studies have shown that the interaction of many factors is necessary for leisure participation, but the ultimate goal is to encourage continued participation [28]. However, even if leisure participants are highly motivated to participate, have successfully removed barriers to participation, and have achieved satisfactory results, it is difficult to ensure continued participation [36,37]. It is inferred that continuous participation intention of the older adults in leisure activities appeared due to the prolonged life expectancy every year [38,39]. Therefore, the results of this study are significant. Specifically, this study analyzed whether exergaming can be a new leisure participation alternative for older adults and found that it has potential. The share of the older adult population in social demographics is increasing daily, and it is necessary to provide physical activity for them. We believe that exergaming, which breaks from the traditional leisure industry and incorporates advancing technology, is a new area that will change the entire leisure industry in the future.

## 5. Conclusions

Technological advances have profoundly affected the leisure industry. New leisure industries with advanced technology are expected to have positive outcomes in terms of reducing leisure constraints and increasing participation rates. In particular, older adults, who are generally considered vulnerable in the leisure industry, have a major opportunity to minimize leisure participation constraints and increase the benefits of leisure participation. Thus, it is necessary to analyze the changing leisure industry with the development of advanced technology for the growing older adult population. This study empirically analyzed whether differences existed in leisure constraints, participation benefits, and continuous participation intention in virtual golf, which can be represented by exergaming, depending on the participants’ age. Specifically, it aimed to determine the effectiveness of technological advances in enhancing physical activity opportunities for older adults. Therefore, the hypotheses that participation constraints, participation benefits, and intention to continue participating in virtual golf (exergaming) would show differences based on age were scientifically verified.

Despite these meaningful findings, this study offers the following suggestions for future research based on the limitations experienced. This study obtained meaningful results regarding how older leisure participants experience participation in virtual golf. Future research should build on the findings of this study and further analyze the experiences of virtual golf participants aged 65 years and older. Specifically, it is necessary to analyze virtual golf participants who have experienced social retirement by segmenting them according to demographic factors. This could be an approach to overcome the research limitations related to the characteristics of the survey participants in this study. Such research attempts are essential to expand current leisure sports with advanced technology, such as VR, beyond golf and to expand leisure participation opportunities among the older adult population.

In addition, this study focused on analyzing the emergence and phenomenon of exergaming from the perspective of the social science field. It attempted to understand leisure participants from sociological, psychological, and management perspectives and found meaningful results. However, future research could design an experimental study from a physiological perspective to determine the extent to which virtual golf positively affects the participants’ physical health. This would serve as an opportunity to derive scientific results that can be considered a limitation of this study. An experimental study with older adult participants divided into experimental and control groups would be a meaningful research endeavor that would expand on the results of this study. Such research will be an important resource for anticipating and preparing the leisure industry, which will rapidly change with the development of technology in the future.

## Figures and Tables

**Table 1 healthcare-12-00962-t001:** Descriptive statistics of survey respondents.

		Group 1	Group 2	Group 3
		Young Adults	Middle-Aged Adults	Older Adults
Gender	Male	36 (32.1%)	66 (60.0%)	58 (65.9%)
	Female	76 (67.9%)	44 (40.0%)	30 (34.1%)
Monthly income (KRW)	Less than 1,000,000	13 (11.6%)	2 (1.8%)	8 (9.1%)
	1,000,000–2,999,999	33 (29.5%)	15 (13.6%)	19 (21.6%)
	3,000,000–4,999,999	49 (43.8%)	57 (51.8%)	29 (33.0%)
	More than 5,000,000	17 (15.2%)	36 (32.7%)	32 (36.4%)
Golf experience (years)	Less than 1	50 (44.6%)	20 (18.2%)	11 (12.5%)
	1–4	56 (50.0%)	53 (48.2%)	26 (29.5%)
	5–9	5 (4.5%)	21 (19.1%)	21 (23.9%)
	10–19	1 (0.9%)	14 (12.7%)	18 (20.5%)
	More than 20	-	2 (1.8%)	12 (13.6%)
Golf mastery (handicap)	Less than 9	19 (17.0%)	13 (11.8%)	5 (5.7%)
	10–19	46 (41.1%)	50 (45.5%)	32 (36.4%)
	20–29	16 (14.2%)	24 (21.8%)	23 (26.1%)
	More than 30	1 (0.9%)	4 (3.6%)	16 (18.2%)
	I do not know	30 (26.8%)	19 (17.3%)	12 (13.6%)
Frequency of playing virtual golf	Once a month	42 (37.5%)	43 (39.1%)	36 (40.9%)
	2–3 times per month	36 (32.1%)	37 (33.6%)	39 (44.2%)
	Once a week	18 (16.1%)	15 (13.6%)	11 (12.5%)
	2–3 times per month	16 (14.3%)	14 (12.7%)	1 (1.1%)
	Almost every day	-	1 (0.9%)	1 (1.1%)
Total		112 (100.0%)	110 (100/0%)	88 (100/0%)

Note. % = Amount in each hundred in each group.

**Table 2 healthcare-12-00962-t002:** Results of validity and reliability.

Items of Participation Constraints Factor	1	2	3	4
(Cost)				
I don’t have enough money to play virtual golf.	0.897	0.005	−0.032	0.102
The fee for entry is too expensive.	0.893	−0.046	0.001	0.034
I cannot afford the fee for entry.	0.865	−0.043	0.024	0.127
(Health)				
I am not fit enough to play virtual golf.	0.000	0.908	0.145	0.042
I don’t have the energy to play virtual golf.	−0.025	0.869	0.169	0.021
Health problems prevent me from playing virtual golf.	−0.064	0.816	0.211	0.056
(Confidence)				
I am not experienced.	−0.017	0.096	0.916	−0.042
My game is too inconsistent.	−0.071	0.211	0.863	0.043
The game is too difficult.	0.083	0.236	0.811	0.010
(Social)				
My friends/family don’t want me to play virtual golf.	0.093	0.037	0.025	0.914
My friends have different interests other than virtual golf.	0.039	0.058	0.036	0.891
I don’t have friends to play virtual golf with.	0.121	0.019	−0.047	0.789
Eigenvalues	3.300	2.784	1.899	1.404
Variance (%)	27.498	23.200	15.829	11.697
Cronbach’s alpha	0.868	0.849	0.857	0.841
Items of participation benefits factor	1	2	3	4
(Personal satisfaction)				
I get good stimulation through virtual golf in daily life.	0.875	−0.026	0.283	0.045
I get a chance to realize myself.	0.864	0.078	0.236	−0.003
I feel a lot of freedom through virtual golf.	0.852	−0.031	0.314	0.009
(Physical)				
Virtual golf helps maintain physical health.	−0.060	0.921	0.057	0.000
My stamina is improving through virtual golf.	−0.028	0.913	0.040	−0.051
I gain physical vitality through virtual golf.	0.105	0.809	−0.022	0.057
(Social)				
Virtual golf helps me maintain relationships with people.	0.284	0.040	0.850	−0.013
Virtual golf helps me promote friendship.	0.311	0.024	0.846	0.007
Virtual golf helps me overcome social alienation.	0.211	0.018	0.792	0.070
(Mental)				
Virtual golf brings about positive changes in emotions.	0.004	0.030	0.045	0.916
I feel a lot of happiness through virtual golf.	−0.014	0.025	0.105	0.892
I relieve stress through virtual golf.	0.048	−0.038	−0.074	0.783
Eigenvalues	3.813	2.344	2.244	1.014
Variance (%)	31.776	19.534	18.696	8.448
Cronbach’s alpha	0.894	0.856	0.840	0.833

**Table 3 healthcare-12-00962-t003:** Results of multivariate analysis of variance.

Variables	Sub-Factors	*df*	*F*	*p*	*η* ^2^		Mean		
							G1	G2	G3
Participation constraints	Health	2	0.551	0.577	0.004	-	1.399	1.427	1.489
	Confidence	2	1.924	0.148	0.012	-	2.360	2.203	2.121
	Social	2	1.432	0.240	0.009	-	3.470	3.279	3.367
	Cost	2	14.949	0.000 ***	0.089	a > b, c	3.708	3.200	3.148
Participation benefits	Physical	2	9.346	0.000 ***	0.057	b < c	3.446	3.203	3.663
	Mental	2	7.249	0.000 ***	0.045	a, b < c	3.393	3.379	3.742
	Social	2	0.517	0.597	0.003	-	3.637	3.552	3.648
	Personal satisfaction	2	3.673	0.027	0.023	-	3.179	3.433	3.420
Continuous participation intention		2	6.486	0.002 **	0.041	a < c	3.625	3.836	4.011

Note. *** *p* < 0.001, ** *p* < 0.01. a = Young adult group (G1); b = Middle-aged adult group (G2); c = Older adult group (G3).

**Table 4 healthcare-12-00962-t004:** Results of post-hoc analyses.

		Participation Constraints				Participation Benefits				Continuous Participation Intention
		Health	Confidence	Social	Cost	Physical	Mental	Social	Personal Satisfaction	
G1	G2	0.941	0.417	0.241	0.000 ***	0.054	0.990	0.699	0.052	0.118
	G3	0.583	0.167	0.694	0.000 ***	0.129	0.005 **	0.995	0.094	0.002 **
G2	G1	0.941	0.417	0.241	0.000 ***	0.054	0.990	0.699	0.052	0.118
	G3	0.779	0.811	0.764	0.907	0.000 ***	0.003 **	0.670	0.993	0.274
G3	G1	0.583	0.167	0.694	0.000 ***	0.129	0.005 **	0.995	0.094	0.002
	G2	0.779	0.811	0.764	0.907	0.000 ***	0.003 **	0.670	0.993	0.274

Note. G1 = Young adult group; G2 = Middle-aged adult group; G3 = Older adult group. *** *p* < 0.001, ** *p* < 0.01.

## Data Availability

Further inquiries beyond the data presented in this paper can be directed to the corresponding authors.

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
