# Peer review of "Virtual Golf, “Exergaming”, Using Virtual Reality for Healthcare in Older Adults: Focusing on Leisure Constraints, Participation Benefits, and Continuous Participation Intention"

_healthcare, 2024, doi:10.3390/healthcare12100962_

Round 1

Reviewer 1 Report

Comments and Suggestions for Authors

Dear authors,

The manuscript is interesting and reports important information. However, I must address the issues below:

- The introduction section is way too large. Please, atain to the essential information to provide a rationale for the reader.

- You report that many other studies had already shown the value of using exergames for the same purposes of your study. Spend more of your intro describing the novelty of the current study and also the clinical meaning of your possible findings.

- I could not notice a sample size calculation, nor a post hoc power analysis. Please, clarify and report.

- Report the instrument's psychometric measures, and the score calculation. Is there distinct dimensions in the instrument? If so, please report their subscores and how they are interpreted.

- To use the MANOVA method, you must previously report the asumption checks. Please, do so. Also, you use a post hoc analysis for your pairwise comparisons, but the method was not described in the methods' section.

- There is an entire section for scale validity and realiability, not previously described in your objetives, nor in your methods section. Please, clarify the aim of your study.

- Restructure the discussion to improve readability. The 1st paragraph should summarize the results, followed by a literature confrontation. Next, the authors should state the meaning of the findings and the study's limitations. 

Author Response

The response letter to the reviewer's comments has been attached. 

Reviewer 2 Report

Comments and Suggestions for Authors

I think that in the title itself it should be indicated

the sport "Exergaming" the virtual goal

I think that the specific context in which the research is carried out should appear in the summary.

In this study, the factors used to measure leisure limitations and the continuum.........The participation restriction factor consists of four sub-factors: (a) cost (b) health (c) trust and ( d) social

......

Next, the factor used to understand the benefits of participation in virtual golf in this ....... The benefits of participation also had four subfactors (a) personal satisfaction (b) physical satisfaction (three items), (c ) social satisfaction and (d) mental satisfaction.....

Well, I think that the authors should make their connection with the measurement instruments they use clearer in both the objectives and the hypotheses.

The authors indicate

Hypothesis 1. Participation constraints differ according to the age of virtual golf participants.

Hypothesis 2. Participation gains differ based on the age of virtual golf participants.

Hypothesis 3. Continuous participation intention differs based on the age of virtual golf participants.

Well, this must be contemplated more clearly in the conclusions.

I ask the authors to review these works and include them in their introduction

Gil-Madrona P, Aguilar-Jurado MÁ, Honrubia-Montesinos C, López-Sánchez GF. Physical Activity and Health Habits of 17- to 25-Year-Old Young People during Their Free Time. Sustainability. 2019; 11(23):6577. https://doi.org/10.3390/su11236577

Gil-Madrona, P., Prieto-Ayuso, A., Silva, S. A. D. S., Serra-Olivares, J., Aguilar Jurado, M. Á., y Díaz-Suárez, A. (2019a). Hábitos y comportamientos relacionados con la salud en adolescentes durante su tiempo libre. Anales de Psicología 35, 140–147. doi: 10.6018/analesps.35.1.301611

Author Response

(The authors gave the same response as above.)

Reviewer 3 Report

Comments and Suggestions for Authors

First of all, thank you for the opportunity to evaluate this article. From a technical point of view, the work is correctly done, it corresponds to the criteria for making a valid scientific work.

My only suggestion to the authors is for the authors to explicitly state whether the virtual golf game involves actually making specific physical movements or is played similarly to other mind games such as chess. This clarification is essential in my view to understand whether playing virtual golf is a physical activity or just a mental activity. The difference between these two types of games can influence how they are perceived and played, as well as the benefits they bring to users. If the game of virtual golf involves making specific physical movements, it can provide benefits similar to those of playing the real sport, such as improving coordination, flexibility and physical endurance. On the other hand, if the virtual game focuses exclusively on the mental and strategic aspects of golf, the benefits may be more related to the development of cognitive skills such as concentration, quick decision making and problem solving.

By clearly indicating how virtual golf is played, authors can help users better understand the nature of the activity and expect certain specific benefits. This transparency is important to promote a healthy and conscious practice of virtual golf, as well as to highlight its impact on the physical and mental health of users.

The degree of participants regarding the level of satisfaction is less relevant in the context of the mentioned idea.

Author Response

(The authors gave the same response as above.)

Reviewer 4 Report

Comments and Suggestions for Authors

 Reviewer Comment

I think that the subject of the study is important in terms of prolonging human life and supporting the World Health Organization's action plan on healthy aging. However, it is seen that there are many parts of the manuscript from the abstract to the end that do not comply with the article writing system. Statistical and numerical information is not provided in the results in the summary section, this information should be added. Additionally, the conclusion section of the abstract was not written appropriately, and this section should be revised. The introduction section is written very long. There is a lot of information that is unnecessary and far from the purpose of the subject, and these should be removed from the introduction section. Information regarding the essence of the subject should be emphasized. Additionally, there are many informative definitions in the introduction section. This spelling system is not correct for entry. Identifying information must be provided at the beginning of the entry. Then, it should be stated what research has been done on this subject in the literature and what results have emerged. Then, based on these studies, the justification for the study should be established by revealing the difference of your study, and then this justification should constitute the purpose of the study. The introduction section should be designed in this systematic way. In addition, in some parts of the introduction, quotes are made without citing the source. These sentences must be cited as a source. The method section is written very superficially and contains limited information about the research method. The method should be presented systematically under the headings of study design, participants, data collection tools and data analysis. In the Result section, a statistical analysis should be made to reveal the numerical homogeneity of the groups and this should be added to Table 1. The discussion section is written shorter and superficially than the introduction section. It is unnecessary to provide general information in the introduction part of the discussion. Direct general results should be given in summary form. Additionally, there has not been a sufficient relationship between the research results and the literature. Personal opinions are given a lot of place in the inferences. The entire discussion should be further linked to research in the literature. Inferences and comments should be cited from the literature. No information was given regarding the limitations of the research. Information about the limitations of the study should be added at the end of the discussion. Revision recommendations for Manucript are provided in more detail below. Please make all suggestions regularly and carefully.

Revision

1.      Page 1, line 23; Add the p significance value (such as p = 0.012) and numerical data to the result section of the summary.

2.      Page 1, line 26; “This study provides objective data on the opportunities and benefits of participation in leisure sports for the growing older population.” Inferences and comments about the results should be included in the conclusion section of the summary. The sentence you use here is not a conclusion sentence. Rewrite the Conclusion section.

3.      The initial letters of the keywords should be written in capital letters.

4.      Page1-2, line 31-62; In the introduction, a lot of detailed information about old age and aging statistics is given. This is quite long for the introduction part of the article. The main focus of the topic should be included in the introduction. Please shorten this section and give more concise and concise information about the statistical information.

5.      Page 3, line 101-115; In this section, a paragraph was written without citing any sources. You must cite the source for all the information you provide in the introduction section and in an article. Make the necessary reference.

6.      Page 3, line 120-139; Most of this paragraph contains information without citing the source. In this paragraph, each sentence contains a different scope of information. Separate sources must be cited for all of this information.

7.      Page 4, line 140-148; The purpose of the study was not adequately justified. Justifications should not be made with general information. Descriptive information about the subject should be given at the beginning of the introduction section. Then, research on the subject should be presented. Then, the justification for the research should be stated by stating the difference between your study and these studies. Finally, the introduction section should be concluded by linking the purpose of the study to this section. In this context, make the necessary arrangements within the systematic.

8.      The methods section is very superficial. Much more information should be given. Additionally, information should be presented systematically under subheadings. Organize the method section systematically as indicated below.

Study design: The place and date of the study should be described in this section. A brief plan of the study should also be stated in this section. When reading this part of the study, the plan of the study should be explained.

Participant: It should be explained how many people were included in the study, in what age range, and the inclusion and exclusion criteria. Information about how the sample size for the study was determined, whether power analysis was performed, and if so, how it was performed should be added. A flow diagram should be added to this section as a figure.

Data collection tools: This section should start with what information is collected as sociodemographic data, and then it should be clearly stated which evaluation scale is used. The name of the data collection tool should be clearly stated. It should be stated what the increase and decrease in the score of the survey consists of.

Data analysis: All statistical information used in the study should be stated more clearly. The significance value for p is not specified in this section. It should definitely be added.

9.      A comparison of the descriptive data of the three groups should be made in Table 1. Statistical analysis should be done as a result and the p value should be added. It should be known whether the groups are numerically homogeneous or not.

10.  What the % means should be written clearly under the tables.

11.  Page 9, line 258-265; “Technological advances have profoundly affected the leisure industry. New leisure industries with advanced technology are expected to have positive outcomes in terms of reducing leisure constraints and increasing participation rates. Particularly for older adults, who are generally considered vulnerable in the leisure industry, there is a major opportunity to minimize leisure participation constraints and increase the benefits of leisure participation. It is necessary to analyze the changing leisure industry with the development of advanced technology for the growing older adult population.” This section is general information; the general result of your study should be given briefly at the beginning of the discussion. Delete this information and create a paragraph summarizing the overall results.

12.  Page 9, line 292-293; “Previous studies have documented the physical, mental, and social benefits of participation in leisure sports [28].” Although information is given as previous studies, a single source is cited at the end of the sentence. If multiple sources are mentioned, more than one reference should be used. Make the necessary adjustments.

13.  The discussion was generally not made with the right relationships in place. The results of your study should be better correlated with research in the literature. It also seems that some of your conclusions contain a lot of personal interpretation. Your inferences should be made by citing studies in the literature, away from personal opinions. Apply the necessary arrangements throughout the discussion.

14.  What are the limitations of the study? No information about this is given in the manuscript. This information should be correctly added to the end of the discussion.

Comments on the Quality of English Language

I think that there are meaninglessness in sentence transitions in my manuscript and it should be reconsidered.

Author Response

(The authors gave the same response as above.)

Round 2

Reviewer 1 Report

Comments and Suggestions for Authors

Please, see the page 4. The table was not well formatted.

Author Response

Dear Reviewer, 

The manuscript (including the Table you mentioned) has been re-formatted. 

Thank you for your support.

Reviewer 4 Report

Comments and Suggestions for Authors

Reviewer Comments

It is seen that the manuscript has transformed into a higher quality form with the proposed regulations. However, some changes and adjustments are still required. In general, the manuscript should be reviewed for fluency of writing. In addition, conjunctions that are used repeatedly should be avoided. Inclusion and exclusion criteria were not adequately explained in the participants section. How many people left the study should be stated in this section and a flow diagram should be added to this section. Descriptive data is given in the data collection tools section, this part should be removed from this section and added to the results section. Finally, the discussion should be made a little deeper. The necessary regulations are suggested below.

Revision

1.      Page 1, line83-88; “Therefore, the purpose of this study is to empirically analyze whether there are differences in leisure constraints, participation benefits, and continuous participation intention in virtual golf, which can be represented as exergaming, depending on the age of the participants. In other words, the purpose of this study is to determine the effectiveness of technological advancements in enhancing physical activity among older adults. In other words, this study aimed to scientifically prove the hypothesis that there are differences in participation constraints, participation benefits, and intention to continue participating in exergaming (virtual golf) based on age.” The word "in other words" has been used frequently one after the other. Use in this way disrupts the reading flow. Write the purpose of the study holistically, without repetition and by rearranging it with words that will not disrupt the flow.

2.      Page 3, line 102; Add your inclusion and exclusion criteria to the Participants section. You should also add the information about the people excluded from the study to the participants section in the data collection tools section. Also add the flow diagram of the participants to this section.

3.      Page 3, line 111; “Age was used as an independent variable for the comparative analysis in this study. Based on the collected age, information, the survey respondents were divided into three groups: Group 1, young adults (18-35 years); Group 2, middle-112 aged adults (36-55 years); and Group 3, older adults (56-69 years). Detailed sociodemographic information of the survey respondents is presented in Table 1.” Move the descriptive information table provided in the data collection tools to the results section (table 1). Only the text of the description should remain here and the table should be moved to the result section.

4.      Page 4, line 130; Add interest in the significance value of p to the data analysis section.

5.      Page 7, line 200; The discussion section needs to be expanded further. The results of the research should be discussed in more depth by relating them to the literature. All inferences should be made with literature support.

Comments on the Quality of English Language

I think that the English writing of the manuscript is not very good and should be reconsidered.

Author Response

Thank you for your comments. 

Based on the comments, the manuscript has been revised. 
